# Oral Supplementation with Benzylamine Delays the Onset of Diabetes in Obese and Diabetic db-/- Mice

**DOI:** 10.3390/nu13082622

**Published:** 2021-07-29

**Authors:** Zsuzsa Iffiú-Soltesz, Estelle Wanecq, László Tóthfalusi, Éva Szökő, Christian Carpéné

**Affiliations:** 1Institut des Maladies Métaboliques et Cardiovasculaires (I2MC), Institut National de la Santé et de la Recherche Médicale (INSERM U1048), I2MC, CEDEX 4, 31432 Toulouse, France; zsuzsaiffiusoltesz@gmail.com (Z.I.-S.); christian.carpene@univ-tlse3.fr (E.W.); 2I2MC, University of Toulouse, UMR1048, Paul Sabatier University, 31432 Toulouse, France; 3Department of Pharmacodynamics, Semmelweis University, 1085 Budapest, Hungary; tothfalusi.laszlo@pharma.semmelweis-univ.hu (L.T.); szoko.eva@pharma.semmelweis-univ.hu (É.S.)

**Keywords:** semicarbazide-sensitive amine oxidase, vascular adhesion protein, copper containing amine oxidase, insulin-resistant diabetes, glycemia, glucose transport, hydrogen peroxide, adipocytes, dietary amines, antihyperglycemic phytochemicals

## Abstract

Substrates of semicarbazide-sensitive amine oxidase (SSAO) exert insulin-like actions in adipocytes. One of them, benzylamine (Bza) exhibits antihyperglycemic properties in several rodent models of diabetes. To further study the antidiabetic potential of this naturally occurring amine, a model of severe type 2 diabetes, the obese db-/- mouse, was subjected to oral Bza administration. To this end, db-/- mice and their lean littermates were treated at 4 weeks of age by adding 0.5% Bza in drinking water for seven weeks. Body mass, fat content, blood glucose and urinary glucose output were followed while adipocyte insulin responsiveness and gene expression were checked at the end of supplementation, together with aorta nitrites. Bza supplementation delayed the appearance of hyperglycemia, abolished polydypsia and glycosuria in obese/diabetic mice without any detectable effect in lean control, except for a reduction in food intake observed in both genotypes. The improvement of glucose homeostasis was observed in db-/- mice at the expense of increased fat deposition, especially in the subcutaneous white adipose tissue (SCWAT), without sign of worsened inflammation or insulin responsiveness and with lowered circulating triglycerides and uric acid, while NO bioavailability was increased in aorta. The higher capacity of SSAO in oxidizing Bza in SCWAT, found in the obese mice, was unaltered by Bza supplementation and likely involved in the activation of glucose utilization by adipocytes. We propose that Bza oxidation in tissues, which produces hydrogen peroxide mainly in SCWAT, facilitates insulin-independent glucose utilization. Bza could be considered as a potential agent for dietary supplementation aiming at preventing diabetic complications.

## 1. Introduction

Benzylamine (Bza) is a small molecule consisting of an aromatic ring, which is organic and non-polar in character (benzyl), with a CH_2_-NH_2_ (amine) highly hydrophilic group. This molecule is synthetically created for multi-purpose uses in organic chemistry since it is a useful chemical intermediate for the synthesis of dyes, pharmaceuticals and polymers in industrial manufacturing. Bza, which possess excellent CO_2_ absorbent properties [1], is also a naturally occurring amine present in foods alongside more widely known dietary amines (tyramine, histamine, tryptamine…). Indeed, Bza is one of the simplest natural alkaloids, belonging to the subclass of protoalkaloids, having their nitrogen outside the aromatic cycle. Accordingly, Bza is found a wide variety of plants: it is present in edible vegetables and has been detected at doses varying from 1 to 10 mg/kg in fresh cabbages, salads, carrots, radishes, as well as in fruits [2,3]. Moreover, the production of Bza by the intestinal microbiota has been proposed to occur during the digestion of edible cruciferous plants [4]. In addition, various substituted benzylamine forms are present in shrubs, such as 2-acetyl-benzylamine in *Adhatoda vasica* [5], 4-hydroxybenzylamine in *Cydonia vulgaris*, [6], or tribenzylamine in hops [7]. Bza has also been evidenced in diverse medicinal plants, such as *Lepidium meyenii* (Maca) [8,9] and *Moringa oleifera* (Drumstick tree) [10,11]. Many parts of the tree *Moringa oleifera* are widely used for multiple applications, including human nutrition in developing countries [12], and recognized in folk medicine for their hypoglycemic activity. In the extracts from Moringa that have been certified to lower blood glucose in diabetic animal models [13,14,15], one active principle, called moringine or phenylmethanamine, is identical to Bza. However, these observations are not sufficient to currently include Bza into the list of the most known biogenic amines, since only traces are expected to occur in human tissues, and to the best of our knowledge, have not been quantified so far. This explains why Bza is still used as a derivatization agent [16] or as an internal standard [17] for chromatography analysis of biological amines in tissues. Nevertheless, the observations listed below led us to study the putative benefits of an oral benzylamine supplementation on a model of severe insulin-resistant diabetes, the obese and diabetic db/db mouse.

First, Bza is the reference substrate of semicarbazide-sensitive amine oxidase (SSAO) [18], a membrane-bound enzyme identical to vascular adhesion protein-1 (VAP-1) [19], which was previously called benzylamine oxidase [20], then renamed primary amine oxidase [21] since its substrate specificity and its sensitivity to inhibitors are far from being limited to Bza and semicarbazide, respectively. In fact, this multifunctional enzyme is one of the four mammalian types of copper-containing amine oxidases, encoded by the *AOC3* gene and will be named AOC3 hereafter (see [22,23] for recent reviews).

Second, Bza has been reported to directly exert beneficial effects on glucose handling in diabetic rats, especially when combined with low doses of vanadium and repeatedly injected or continuously administered via implanted osmotic minipumps [24,25]. The suggested underlying mechanism of the antihyperglycemic action of Bza was related to the strong stimulation of glucose uptake it promotes in rat fat cells in the presence of vanadium [26,27]. Such insulin-like effect was reproduced with other AOC3 substrates, and was blocked by amine oxidase inhibitors or by antioxidants [28]. Likewise, it was abolished in mice genetically invalidated for the *Aoc3* gene [29]. Bza-induced stimulation of glucose uptake is mediated by the hydrogen peroxide produced during oxidative deamination, which interacts with vanadium, and generates peroxovanadate, a powerful insulin-mimicking agent [30] that inhibits tyrosine phosphatases [31]. 

Of note, diverse insulin-like effects of Bza have been observed without the need for exogenous vanadium, especially in cultured preadipocytes [32,33,34], mouse adipocytes [35,36], and human adipocytes [37,38]. Moreover, we have shown, by studying Bza diffusion across gut sacs, that the intestinal barrier and its elevated amine oxidase activity are not insurmountable for Bza absorption [39]. Previous pioneering studies have established that in humans, 98 % of the ingested Bza is found as hippuric acid in the urine within 24 h [40]. Since a substantial proportion of ingested Bza is metabolized, the idea that oral administration of Bza can reach sufficient tissue levels to reproduce the insulin-like actions observed in vitro in fat cells prompted us to perform chronic oral Bza supplementation in animal models. Considering that we have already reported that prolonged oral administration of 0.4% Bza solution mitigates several of the disturbances induced by high-fat feeding in mice [39], we investigated in the present study whether oral Bza supplementation for seven weeks could exert antidiabetic effects in a model of severe insulin-resistant diabetes: the obese and diabetic db/db mouse (C57BL/KsJ strain). 

Bearing a recessive mutation for the leptin receptor (db-/-), this widely recognized murine model of diabetes becomes rapidly hyperphagic and exhibits a dramatic increase in blood glucose levels after weaning [41,42]. Since we observed that AOC3 expression is considerably increased in the fat depots of the db-/- mice relative to lean control [43], we supposed that comparing the effect of a Bza suplementation in young obese and lean littermates could bring additional convincing observations about the potential antidiabetic action of orally given Bza.

## 2. Materials and Methods

### 2.1. Animal Model

Mice with a naturally occurring mutation of the *db* gene that makes them deficient in leptin receptor were handled in accordance with the European Community Council Directives for experimental animal care. These C57BL/KsJ mice were obtained from Charles River Laboratories France and housed at 22 °C with a 12-h light/dark cycle and were fed *ad libitum* as already described [44]. Heterozygous for the mutation of *db* gene, the db+/- mice were continuously interbred to provide all three genotypes: obese and diabetic db-/-, normoglycemic db+/-, and lean db+/+. 

### 2.2. Group Composition and Oral Benzylamine Supplementation

After weaning at four weeks of age, the littermates (3–15 mice per litter) were separated by gender and the litters were further divided in two groups, one receiving 0.5% benzylamine hydrochloride (Bza) in the drinking water (2 g Bza per 400 mL tap water, changed weekly), the other remaining untreated. Body mass was measured weekly, and non-fasting blood glucose twice a week until the end of the treatment. At the age of six weeks, the db-/- mice, were overweight since they are recognized to exhibit early onset of obesity and hyperglycemia [44], and were separated from their lean db+/- littermates to facilitate the weekly follow-up of food and liquid consumption. The db+/+ offspring were discarded from the experiment on the basis of their smaller body weight and the color of their fur. This means that for a period between four and six week of age, data were collected without knowing whether the littermates would become obese or not. The complete age-dependent curves for body weight and non-fasting glycemia were thus reconstituted *post hoc* as soon as week 4 for all the groups. Although each genotype was obtained with the expected Mendelian frequency, it was practically difficult to perform a complete study with exactly the same number of animals for each group and for their age- and sex-matched littermates in all the planned in vivo explorations and in vitro experiments. The number of studied animals will therefore be indicated as *n* in each subset of investigation together with the male/female ratio.

### 2.3. Non-Invasive Explorations

During the supplementation, mice were randomly chosen to undergo one of the following non-invasive measurements, after the required trials for their adaptation: Magnetic Resonance Imaging scans (MRI), using Echo MRI NMR device (100TM3; Echo Medical Systems, Houston, TX, USA) for total fat content determination during 1-min measurements as previously detailed [45], or for daily urine emission, by collecting feces and urines in individual glass metabolic cages for 24 h.

### 2.4. Tissue Sampling and Determination of Blood Biochemistry

Glucose level was determined twice a week on a drop of blood collected from the tail vein in non-fasting state (between 09:30 and 10:30), by using the glucose oxidase method (MPR 3 glucose/GOD-PAP, from Boehringer, Mannheim, Germany). At the end of the treatment, the intra-abdominal (INWAT) and subcutaneous (SCWAT) white adipose tissues were removed, weighed and frozen for further analyses with the exception of perigonadal fat pads used for the preparation of freshly isolated adipocytes, and a portion of inguinal SCWAT used for immediate determination of hydrogen peroxide release. Liver and aorta were frozen at −80 °C. On plasma samples obtained at euthanasia after overnight fasting, immunoreactive insulin (IRI) was determined with an insulin RIA kit (Linco Research, Inc., St. Charles, MO, USA) and other plasma parameters were measured using kits following manufacturer’s instructions as already described [46].

### 2.5. Adipocyte Preparation, Lipolysis and Glucose Transport Measurements

Adipocytes were isolated from perigonadal fat pads by using liberase at 0.08 U/mL in Krebs-Ringer buffer containing bicarbonate, HEPES and 3.5% bovine serum albumin. Lipolytic activity was assessed by the glycerol released during 90-min incubation with the tested agents [47], while [^3^H]-2-D-deoxyglucose (2-DG) uptake measurements were performed as already detailed [38].

### 2.6. RNA Extraction and Quantitative Real-Time RT-PCR

400 mg of WAT or liver were homogenized in 2 mL of a denaturing guanidine-thiocyanate-containing buffer supplied by the RNeasy mini kit (Qiagen, Courtaboeuf, France) plus 20 μL β-mercaptoethanol. After lipid elimination by chloroform extraction, total RNA was extracted according to the supplier’s instructions. Then, 0.5 μg of total RNA was reverse-transcribed using random hexamers and Superscript II reverse transcriptase (Invitrogen, Cergy Pontoise, France). Real-time PCR was performed using Sybr Green TaqMan Universal PCR Mastermix (Eurogentec, Angers, France) and the GeneAmp 7500 detection system instrument (Applied Biosystems, Foster City, CA, USA) according to the protocols provided by the manufacturers. The relative mRNA transcript levels were calculated according to the 2^−ΔΔCt^ method as detailed [48]. Oligonucleotide primers were designed using Primer Express (Perkin-Elmer Life Sciences, Courtaboeuf, France) and verified on Blast Nucleotide software (National Center for Biotechnology Information, Bethesda, MD, USA). The sequences of validated primers are given in Table 1 or available in [45,46].

### 2.7. Assessment of H_2_O_2_ Release and of NO Bioavailability

Hydrogen peroxide was detected owing to the use of a peroxidase-based fluorometric method. Small pieces of freshly removed, intact WAT (c.a. 20 mg) were put in 580 μL phosphate buffer for ten minutes. 20 μL of fluorogenic mixture (40 μmol/L Amplex Red and 4 U/mL horseradish peroxidase) was added at time 0. Then, the plastic tubes were protected from light and incubated for 30 min at 37 °C. 200 μL of the solution in which were floating the WAT pieces was transferred into 96-well dark plates for the detection of fluorescent signal (530/590 nm) by a Fluoroscan Ascent FL plate reader (Thermo Electron Corporation, Vantaa, Finland), as previously described [43]. Hydrogen peroxide gave a linear calibration curve in a concentration range of 0–5 μmol/L. The same hydrogen peroxide determination was used to assess amine oxidase activity in adipose tissue homogenates. Briefly, 50 µL of homogenates were preincubated for 10 min at 37 °C with 50 µL phosphate buffer containing or not inhibitors: 1 mmol/L semicarbazide to inhibit AOC3, 10 µmol/L pargyline to inhibit monoamine oxidase (MAO). Then, 30-min reaction was started by the addition of 50 μL fluorogenic mixture and 50 μL of Bza dilutions (0 to 1 mmol/L). Aorta nitrate and nitrite concentrations were measured by a previously reported method, based on capillary electrophoresis [49].

### 2.8. Reagents

2-[1, 2-^3^H]-D-deoxyglucose was from Perkin Elmer (Waltham, MA, USA). Semicarbazide, pargyline, benzylamine (all hydrochloride salts), horseradish peroxidase, bovine albumin and other chemicals were purchased from Merck-Sigma-Aldrich (St. Quentin Fallavier, France). Liberase and enzymes for glycerol assay were obtained from Roche (Mannheim, Germany). Amplex Red fluorescent dye was from Fluoroprobes/Interchim (Montluçon, France).

### 2.9. Statistical Analyses

The analyses were conducted using Splus 6.1 software version (Insightful, WA, USA). The linear mixed model was fitted to the data collected constitutively during the treatment period to describe time-course changes of blood glucose, body weight, food and water consumption. The initial model contained gender, treatment and genotype as fixed factors and it was assumed that these factors could modify the individual slopes and intercepts of the animals. Initial analysis revealed that gender was clearly a non-significant factor but at the same time that there was highly significant triple interaction between time (age), treatment and genotype, regardless of the dependent variable analyzed: blood glucose, water consumption or body weight. Results are given as means ± SEM. For end-point analyses, two-way ANOVA or Student’s *t* test were used. NS means no significant difference between the compared samples.

## 3. Results

### 3.1. Oral Supplementation of Benzylamine (Bza-Drinking) Limits the Hyperglycemia of db-/- Mice

After weaning, littermates from the crossing of heterozygous genitors were separated by gender and the litters were further divided in two groups, one receiving 0.5 % benzylamine in the drinking water (Bza-drinking), the other remaining untreated (control). Since there was no gender difference in blood glucose levels, Figure 1 summarizes the pattern of glucose levels irrespective of the gender. The blood glucose level of db+/- did not change with age, while the non-fasting glycemia of db-/- mice was higher than that of db+/- and increased from 4 to 11 weeks of age (Figure 1a). In untreated db-/- mice, the non-fasting blood glucose reached 550 mg/100 mL at 11 weeks of age, whilst this increase was delayed and limited to 400 mg/100 mL in Bza-drinking db-/-. Bza supplementation did not alter the normoglycemia of db+/- (Figure 1a). Applying the linear mixed statistical model to a total of more than one thousand observations obtained from 50 cages containing between 1 and 5 littermates, the slope for age-dependent increase of blood glucose was: 52.61 ± 3.90 in db-/- vs. 33.81 ± 3.14 in Bza-drinking db-/- (*p* < 0.0005). Determination of fasting blood glucose at the end of experiment confirmed that Bza-drinking did not have any influence on db+/- mice while it lowered by 42% the typical hyperglycemia of db-/- mice (Figure 1b).

### 3.2. Daily Urine Production and Glucose Urinary Output

Since blood glucose can be lowered either by decreasing hepatic production or dietary intake, by increasing peripheral utilization or even by elimination in urines, the daily urine excretion was measured in individual metabolic cages at four time points during the treatment. The amount of urine collected per 24 h increased with age in db-/-, while this parameter remained constant at a lower level in db+/- mice (Figure 2a).

Such increase in daily urine production did not show gender-specific difference (and was prevented by Bza-drinking in both male and female diabetic animals. Although the Bza-drinking db-/- mice excreted more urine than the normoglycemic db+/- ones, they produced significantly less than the untreated diabetic mice during the last two weeks of treatment. More importantly, the glucose elimination in urines was significantly reduced in Bza-drinking db-/- mice (Figure 2b), although not totally normalized since there was not any glycosuria in untreated db+/- and in Bza-drinking db+/- mice. Thus, the improvement of glucose handling by oral benzylamine supplementation was apparently not mediated by a diuretic effect or by a worsened renal function leading to an elevated glucose output.

### 3.3. Water Consumption

In the db-/- group, drinking behaviour increased with age, reaching the typical polydipsic feature of the type 2 diabetes that takes place in this model. Water intake remained constant in the db+/- littermates (Figure 3). Bza supplementation did not affect water intake in db+/- while it dramatically lowered it in db-/- mice. For statistical analysis, raw data of 141 determinations were fitted into linear mixed model, and resulting regression lines were tested for difference between slopes: 3.72 ± 0.64 in db-/- vs. 0.96 ± 0.26 in Bza-drinking db-/-, traducing a significant improvement at *p* < 0.0005.

### 3.4. Body Weight Gain and Food Intake

Figure 4 shows the growth and food intake in lean and obese groups, regardless of gender. The body mass of 4-week old mice was similar in treated and their respective control groups. As expected, the db-/- mice became frankly obese as a consequence of an impressive hyperphagia. Bza-drinking lean mice did not exhibit any change in their growth when compared to lean control (Figure 4a), with a similar slope for db+/- (1.09 ± 0.07) and Bza-drinking db+/- (1.09 ± 0.07; NS) in spite of a small reduction of the food intake during the last two weeks of treatment (Figure 4b). During the same period, the Bza-drinking db-/- obese mice reduced their food consumption by almost 15 % when compared to untreated db-/-, but this did not limit their growth significantly. In fact, the body mass of untreated obese mice tended to reach a plateau between 9 and 10 weeks of age and even decreased slightly at week 11, whereas Bza-drinking db-/- mice were still growing.

These moderate and late changes participated in differentiating the overall growth of untreated diabetic/obese mice from that of Bza-drinking group: the slope of linear mixed statistical model was 1.84 ± 0.15 for db-/- vs. 2.92 ± 0.18 for Bza-drinking db-/- (*p* < 0.0005). Accordingly, both the final body mass after an overnight fasting and the body weight gain during treatment were higher in the Bza-drinking db-/- than in the obese/diabetic untreated group (Table 2). Therefore, Bza oral administration reduced the cumulative food intake in both genotypes but seemed to prevent the spontaneous decrease of food efficiency occurring in adult db-/- mice.

Taking into account the age-dependent changes in body mass and in water consumption, it was calculated that the daily intake of Bza was varying, from the beginning to the end of the treatment, between 5100 and 6300 µmol/kg bw/day in lean mice, and between 9300 and 10,100 µmol/kg bw/day in obese mice.

### 3.5. Adiposity and Plasma Markers of Metabolic Disturbances

Adiposity was first assessed by magnetic resonance exploration one week before the end of treatment. This exploration, which could not be performed to all the studied mice in reason of a limited access to the MRI devices, indicated an increased percentage of fat mass in Bza-drinking obese mice (44.3 ± 1.4% in db-/- vs. 49.6 ± 1.6% in Bza-drinking db-/-, *n* = 6 + 7 and 3 + 6 males and females, respectively, *p* < 0.03). Consistent with this increased fattening, the SCWAT mass was clearly larger in Bza-drinking db-/- mice than in the untreated obese/diabetic group (Table 2), when determined at the end of experiment. As expected, INWAT and SCWAT were hypertrophied in db-/- when compared to db+/- littermates, but Bza-drinking did not increase adiposity in the lean mice.

Accordingly, the adiposomatic index, calculated as the percentage of the dissected mass of fat depots relative to final body mass, revealed an increase of fat deposition in the Bza-drinking db-/- when compared to untreated obese mice (15.3 ± 0.5% vs. 13.7 ± 0.4%, *n* = 18 and 28, respectively, *p* < 0.02), while Bza treatment did not modify this index in lean mice. The liver mass, which was increased in db-/- mice, was not normalized after Bza treatment and remained unchanged in the lean group (Table 2).

Fasting plasma parameters confirmed that the onset of diabetes and obesity in db-/- mice was accompanied with disturbances in circulating lipids, such as increased total cholesterol and free fatty acids (FFA). Bza-treatment reduced triglyceridemia but not FFA or cholesterol in the db-/- group and did not affect any of these parameters in db+/- mice (Table 3). The levels of glucose and insulin were impressively elevated in the obese/diabetic mice when compared to lean littermates. The Bza-induced decrease in fasting plasma glucose found in obese mice was accompanied by a decrease in insulin resistance index (attested by relative murine HOMA-IR), while circulating insulin was not altered. A tendency to decrease fructosamine, a surrogate marker of hyperglycemia in mouse was also observed after Bza treatment. Consistently, Bza treatment lowered the elevated uric acid levels of db-/-, while it did not alter those of the lean littermates.

Since Bza had no effect on glycemia, body weight, and plasma metabolic parameters of lean and normoglycemic mice, while it clearly improved the diabetic but not the obese state of db-/- mice, further investigations aimed at studying the influence of Bza supplementation on insulin sensitivity were performed in adipocytes from db-/- mice only.

### 3.6. Glucose Uptake and Lipolytic Activity in Adipocytes from Obese Mice

The insulin responsiveness was explored in adipocytes freshly isolated from perigonadal WAT, by measuring glucose uptake and lipolysis.

In accordance with the insulin-resistant state of db-/- mice, the 2-deoxyglucose transport (2-DG) was poorly activated by insulin in adipocytes from obese mice, since 100 nmol/L insulin induced only a 2- to 3-fold stimulation over basal uptake, while it reached 5-fold in lean heterozygous mice. Indeed, oral Bza treatment did not modify basal 2-DG uptake (0.47 ± 0.09 vs. 0.35 ± 0.05 nmol 2-DG/100 mg lipid/10 min, NS) and tended to increase the insulin-stimulated hexose uptake in adipocytes from obese mice (Figure 5a).

Bza treatment improved the adipocyte sensitivity to insulin since response to 10 nmol/L was equivalent to 56 ± 2% of maximal insulin activation in treated vs. 37 ± 7% in untreated obese mice (*p* < 0.05). In the same conditions, the combination of 1 mmol/L benzylamine plus 0.1 mmol/L vanadium stimulated 2-DG uptake up to 66% and 44% of maximal insulin effect in Bza-treated and untreated db-/-, respectively.

Basal lipolytic activity was unaltered after oral Bza treatment (0.63 ± 0.05 vs. 0.55 ± 0.11 µmol glycerol/100 mg lipid/90 min, NS), as it was the case for the dose-dependent stimulation by the β-adrenergic agonist isoprenaline (Figure 5b).

The antilipolytic action of insulin was tested as the capacity to inhibit the lipolysis promoted by 10 nmol/L isoprenaline: adipocytes from Bza-treated mice exhibited a greater antilipolytic response to 100 nmol/L insulin than did the untreated db-/- (Figure 5c). As expected, benzylamine was also antilipolytic since it partially inhibited isoprenaline-evoked lipolysis, regardless of the presence of 0.1 mmol/L vanadium in the medium (Figure 5d).

Taken together, these data confirmed that Bza directly mimicked several insulin actions when added in vitro to adipocyte preparations, albeit it was unable to totally reverse the insulin resistance of adipocytes from db-/- obese mice when chronically administered in vivo.

### 3.7. Metabolism- and Inflammation-Related Genes in Adipose Tissue and Liver

As the SCWAT mass was clearly increased by Bza-treatment, gene expression was investigated in this fat depot in an attempt to further explore the mechanisms of action of Bza. However, the expression of adipokines and of other genes known to be elevated in the db-/- mouse was far from being normalized after Bza treatment in this tissue; some of them being reported in Table 4. This was the case for the proinflammatory PAI-1 and TNFα, for the endothelial markers involved in vascularisation (CD31, Tie2) and for the AOC3 gene, which encodes for VAP-1/SSAO, and which is considered as a late marker of adipocyte differentiation [22,43,50]. Similarly, the expression of the beneficial adipokine adiponectin was not modified by Bza-treatment. However, the expression the macrophage marker F4/80 tended to be lowered in SCWAT of Bza-drinking db-/-, as it was also the case for of apelin, an adipokine that is elevated in obese states [44] (Figure 6). Anyhow, the increased fat deposition of SCWAT found after Bza supplementation in obese mice was not clearly deteriorating or improving the expression of various genes altered in obese/diabetic db-/- mice.

It was verified whether the increased lipid deposition found in WAT was concomitant with enhanced hepatic lipogenesis. The expression of sterol regulatory element-binding protein 1 (SREBP-1c) and of Fatty acid synthase (FASn), which was exaggerated in the liver of db-/- mice, was almost normalized after Bza supplementation (Figure 6).

Together with the unchanged liver mass, this suggested, that continuous ingestion of the amine lowered blood glucose at the expense of enlarged subcutaneous fat mass but without worsening insulin responsiveness and without promoting hepatic steatosis.

### 3.8. Oxidative Stress Markers in Adipose Tissue and Aorta NO Bioavailability

Finally, it was verified whether Bza administration on a long period induces adverse effects such as oxidative stress, a deleterious outcome that can be suspected upon sustained activation of AOC3, which is a hydrogen peroxide-producing enzyme that might alter the NO bioavailability in blood vessels.

Table 5 shows that the spontaneous hydrogen peroxide release by SWAT homogenates was unchanged after Bza-drinking regardless of the genotype. On the opposite, the AOC3-mediated production of hydrogen peroxide in response to the addition of 0.1 mmol/L benzylamine was higher in obese than in lean mice, and this remained unaltered after prolonged Bza-treatment.

To evaluate the hydrogen peroxide release by SCWAT under conditions closer to the in vivo physiological situation, freshly removed small pieces of fat pads (explants without any homogenization or washing process) were incubated with a fluorogenic mixture that allowed continuous monitoring. There was no change in the ex vivo production of H_2_O_2_ after Bza-treatment in obese mice: 0.46 ± 0.19 vs. 0.46 ± 0.21 pmol hydrogen peroxide/mg wet tissue/min (*n* = 5, NS).

The apparent obesity-related increase in the oxidation of 0.1 mmol/L Bza by adipose depots was confirmed by enzyme kinetics of AOC3 activity, since the V_max_ for the oxidation of increasing doses of benzylamine was, when expressed as pmoles of produced H_2_O_2_/mg protein/min, 212 ± 22 for SCWAT from db+/- and 440 + 72 for SCWAT from db-/- mice (*n* = 7, *p* < 0.05). In contrast, K_m_ values did not differ significantly between the two genotypes (ranging between 22 and 33 µmol/L, NS).

However, on a physiological point of view, the amount of lipids increases much more than the protein content in the WAT of obese mice, when compared to lean. Taking into account that the adipose depots were even more ‘fatty’ in obese than in lean mice, and that their relative proportion of proteins/lipids was reduced, the resulting AOC3 activity per wet tissue mass unit was lower in the obese than in the lean mice: 0.45 ± 0.19 and 1.84 ± 0.51 pmoles of H_2_O_2_ produced/mg inguinal adipose tissue/min, respectively (*n* = 4, *p* < 0.05). Notwithstanding, the almost tenfold larger fat depots in the obese/diabetic than in the lean mice implied that WAT was the major organ involved in Bza oxidation in db-/- mouse, as already reported [43], and this was not altered by Bza supplementation.

In the aorta, nitrite concentration was measured as an index of nitric oxide (NO) availability. It was decreased in db-/- compared to db+/- littermates, while nitrate/nitrite ratio was increased. Bza supplementation restored aortic nitrite levels (Figure 7a), and tended to reduce nitrate/nitrite ratio (Figure 7b). Since nitrite concentration was higher in Bza-drinking obese mice than in the untreated ones, it can be suggested that an enhanced NO bioavailability occurred in the vasculature of the treated mice. Similarly, when considering that the ratio of nitrate-to-nitrite can be used to assess the oxidative metabolism of NO, its decrease in the treated obese/diabetic mice argued for an increased effectiveness of NO [51], almost normalizing the alteration found in the db-/- mice.

## 4. Discussion

In this study, we have shown that an oral supplementation with Bza delays the onset of diabetes in the db-/- mouse, a model of severe type 2 diabetes. Alongside its antihyperglycemic effect, which was evidenced in both non-fasting and fasting conditions, chronic Bza ingestion reduced other complications of the diabetic and obese db-/- mice, such as hyperphagia, polydipsia, polyuria, glycosuria, excessive levels of circulating triglycerides and uric acid. Bza supplementation also reduced several signs of inflammation in WAT and restored the levels of vascular nitrites, which were altered during the onset of diabetes and obesity in db-/- mice. The mitigation of diabetes induced by a supplementation with the naturally occurring amine was concomitant with a small decrease in food intake and occurred at the expense of an increased fat accumulation in WAT, without worsening its inflammation grade, and without increasing hepatic lipogenic pathway. We propose that at least two actions of the amine that have been already reported independently were likely involved in such mitigation of non-insulin-dependent diabetes. They will be discussed below before mentioning other mechanisms that cannot be excluded.

First, the overall antidiabetic effect of oral Bza supplementation relies with its acute antihyperglycemic effect already observed in rabbits subjected to glucose tolerance test [35]. When infused just before an intravenous glucose load, benzylamine is able to limit the hyperglycemic excursion in conscious rabbits, and it has been reported that semicarbazide treatment blocked such effect [35]. Moreover, a similar antihyperglycemic effect has been confirmed for other benzylamine derivatives in mice, which is abolished in mice invalidated for the *Aoc3* gene, encoding AOC3/SSAO/VAP-1 [46]. While these previous studies reported in vivo short-term effects, an antihyperglycemic effect of long-term Bza treatment is evidenced here for the first time. This latter effect is probably mediated by an increased glucose utilization in peripheral tissues, triggered by an oxidation of the amine in the adipose depots, since: (i) the hypertrophied WAT of db-/- mice is quantitatively a major anatomical location for AOC3 activity [43], (ii) the in vitro oxidation of the amine substrate Bza induces various insulin-like effects in fat cells, including the activation of glucose utilization, lipogenesis, and antilipolysis, which are dependent on the hydrogen peroxide generated during amine oxidation by AOC3, and which are abolished in mice devoid of AOC3 activity [52], (iii) the combination of Bza plus vanadate similarly increased glucose tolerance in Goto-Kakizaki obese and diabetic rats [25]. The increased expression of AOC3 in the SCWAT of db-/- mice, evidenced here by greater *Aoc3* mRNA abundance and Bza oxidation capacity, is consistent with a higher metabolism of Bza in the fat depots of the db-/- mouse when compared to lean control [53]. Such elevated AOC3 activity was not altered by the sustained administration of the amine substrate, making this multifunctional enzyme a potential target for novel treatments of obese/diabetic states. The fact that normoglycemia or adiposity of db+/- mice was not influenced by Bza supplementation fits with such suggested mechanism of action, since the overall amount of AOC3 is lower in the WAT of lean animals [43]. Moreover, this paradigm is in perfect agreement with that raised in a preliminary pilot study, in which glycemia was not influenced by Bza oral treatment in non-obese, non-diabetic rats [54], while the hyperglycemic excursion during glucose tolerance tests was reduced.

The second effect of Bza-drinking was a modest decrease of food intake in db-/- and in db+/- mice as well. This decrease reached significance only after four weeks of supplementation but was consistent with previous findings reporting that the amine’s inhibitory effect on appetite was almost immediate and mediated by potassium channel blockade and subsequent release of biogenic amines in the central nervous system [55,56]. Noteworthy, the hypophagic action of Bza is not mediated via its oxidation by AOC3 since amine oxidase inhibitors increased—instead of inhibiting—such effect [53]. However, the hypophagic effects of Bza have been described only in short-term experiments during behavioral studies in non-obese, non-diabetic rodents [55,56] and no reduction of body weight gain after prolonged treatment has been reported so far. In the db-/- model, it must be noted that Bza-drinking limited food intake even in the presence of a disrupted leptin anorectic signaling. Although the reduced calorie intake may have contributed to the reduction in non-fasting blood glucose, it was likely insufficient for normalizing the excessive body weight gain of the db-/- mice. On the opposite, the better glucose utilization elicited by Bza supplementation, which limited glucose leak in urines, resulted in an increase of food efficiency that was associated with an increased body weight gain in spite of a reduced food intake, especially around the age of eleven weeks, at which time the severe diabetic status of the untreated db-/- renders them highly glycosuric and impairs their impressive body weight gain.

Bza treatment also dramatically diminished the elevated water consumption of db-/- mice. A more limited reduction in drinking behaviour has already been observed in non-diabetic rats receiving a 0.3% Bza solution [54] and in high-fat diet mice receiving a 0.4% Bza solution [39]. However, it is difficult to attest that Bza directly influenced drinking behaviour when considering that the reduction in fluid intake accompanied the antihyperglycemic action of Bza and the almost complete obliteration of glucose leak in the urines of young diabetic db-/- mice. Again, the reduction of polyuria and polydypsia, associated to the improved food efficiency in the treated db-/- mice, suggested an improvement of glucose utilization by Bza-drinking rather than a diuretic effect. Probably both central (AOC3-independent) and peripheral (AOC3-mediated) effects of that amine converged to delay the dysregulation of glucose handling in the db-/- mice during the onset of obesity and diabetes.

The AOC3-mediated oxidation of Bza has been demonstrated to support its insulin-like effects such as activation of glucose uptake and incorporation into lipids in rodents [35,52] and in human adipocytes [37,38]. These insulin-like effects were confirmed here in the fat cells of db-/- mice, and were improved rather than altered in the Bza-drinking group. We therefore suggest that a Bza-induced activation of glucose transport occurred in vivo during Bza supplementation and participated in improving the glucose tolerance of the treated mice. Such beneficial effects on glucose handling were supported by enhanced de novo lipogenesis and activated glucose transport in adipocytes. Both resulted in fat accretion, especially in the SCWAT, rich in AOC3 activity in the db-/- littermates, without worsening hepatic steatosis since liver possess a lower AOC3 activity than WAT [57].

Moreover, the elevated adiposity observed at the end of Bza supplementation in db-/- mice is in line with the direct antilipolytic effect that the amine provokes during acute incubation with adipocytes [37,58,59]. Additionally, the observed increase in adiposity in the Bza-drinking obese mice is in perfect agreement with the capacity of sustained Bza stimulation to promote adipogenesis in various preadipocyte lineages, even in the absence of insulin [34,60,61], in a manner that depends on copper availability [62], since AOC3 is a copper-containing enzyme.

In keeping with this line, the increased adipose mass dissected at the end of Bza supplementation was in complete agreement with the results of non-invasive exploration assessed one week earlier by Echo MRI on a lower number of animals and indicative of larger percentage of fat mass in Bza-drinking mice. Thus, Bza-supplementation was probably facilitating the onset of obesity, while it delayed the severity of diabetes. Diverse results indicated that SCWAT was the main target of Bza supplementation. Even if adipocyte hypertrophy occurred during fat depot extension, it was beneficial for the glucose homeostasis of Bza-drinking db-/- mice, especially when considering that the transplantation of SCWAT into visceral regions improves glucose metabolism/tolerance in mice [63]. In other terms, Bza drinking was increasing the proportion of metabolically active WAT in the obese mice, while limiting the growth of “sick fat”, and this was not accompanied by signs of increased inflammation, while concomitant with a decrease in WAT inflammatory macrophage markers such as F4/80.

It has been already evidenced that the short-term or long-term insulin-like effects of Bza are blocked by AOC3 inhibitors or antioxidants, and could be observed with other amine oxidase substrates, arguing that the end-product of amine oxidation involved in the underlying mechanisms is hydrogen peroxide [28]. The pivotal role played by hydrogen peroxide also explains the synergism between vanadium and benzylamine, regarding the insulin-like effect of AOC3 substrates, since it has to be kept in mind that peroxovanadate, one of the best insulin-mimickers, is generated by chemical interaction between hydrogen peroxide and vanadium [31]. However, while such synergism between Bza and vanadium is impressive in rat, it is absent in human adipocytes [37], and poorly evident in mouse (as reported here for the antilipolytic effects of benzylamine, which were almost similar in the presence or the absence of the transition metal in Figure 5). It is therefore possible that the supplementation of mice with only Bza -without vanadium- can reproduce in vivo a large part of the above-mentioned insulin-like effects. Moreover, Bza acts independently of insulin [25,28], and its lowering effect on blood glucose can likely occur in insulin-resistant states, explaining that it can delay the onset of glucotoxicity in young db-/- mice, without showing clear signs of an improvement in insulin sensitivity. In this view, only a tendency to alleviate insulin resistance was found in the antilipolytic activity of adipocytes from Bza-drinking db-/- mice (while no clear leftward shift occurred for the insulin dose-response curves in Figure 5), and the partial recovery of glucose homeostasis was not accompanied with a normalization of the hyperinsulinemia. Thus, Bza-supplementation can exert some insulin-like effects but is not capable of acting as an insulin-sensitizer treatment.

The decreased nitrite concentrations found in the aorta of diabetic db-/- mice are in line with the literature, showing decreased acetylcholine-induced aortic relaxation in diabetes [64]. The improvement of nitrite concentration in Bza-treated db-/- suggests a restored NO production. Hydrogen peroxide is known to increase eNOS expression, and NO release of endothelial cells [65,66]. Alternatively, the higher aorta nitrite levels can be due to the reduction of hyperglycemia and related oxidative stress, thus improving local endothelial function. Moreover, when considering the tissue nitrate-to-nitrite ratio as an indicator of the conversion of NO to reactive nitrogen species by excessive reactive oxygen species, there was no indication of oxidative stress in the aorta of db-/- mice treated with Bza. It must be mentioned that Bza can also be oxidatively deaminated by reactive aldehydes themselves (such as methylglyoxal produced in part by AOC3-dependent aminoacetone oxidation, which is increased in diabetic plasma [67]). Thus, Bza may limit aldehyde toxicity by acting like a scavenger [68].

Another AOC3-dependent effect of Bza supplementation could rely with the role recently attributed to that multifunctional enzyme regarding the selection of fuels by growing adipocytes [62]. Indeed, activation of AOC3, by supplying either substrates or copper, which is an indispensable cofactor, coordinates changes in fuel availability by increasing cellular processes that use carbohydrates, such as glycolysis, de novo lipogenesis [62]. Among the other biological properties attributed to benzylamine derivatives are some cholesterol-lowering effects [69,70], but apparently Bza supplementation failed to improve cholesterol handling in the obese/diabetic mice. Moreover, as plant studies have indicated that Bza possesses molluscicide properties [10], one can presume that Bza could alter the intestinal microbiota of db-/- mice and thereby modify or correct the dysbiosis that is frequent in the obese and diabetic mice [71]. Lastly, Bza supplementation may have enhanced the beneficial effects of polyphenols potentially present in rodent chow diet, at least when considering their amine oxidase-like activity [72] and their interplay with insulin-signaling [71].

## 5. Conclusions

Taken together, all these observations indicated that Bza supplementation improves the glycemic control of the young obese and diabetic db-/- mice at the expense of a larger fat store development, which occurred without worsening the inflammation state. Such capacity to delay the onset of a severe diabetes in a murine model needs to be improved, and deserves further study on its potential interest in clinical nutrition.

## Figures and Tables

**Figure 1 nutrients-13-02622-f001:**
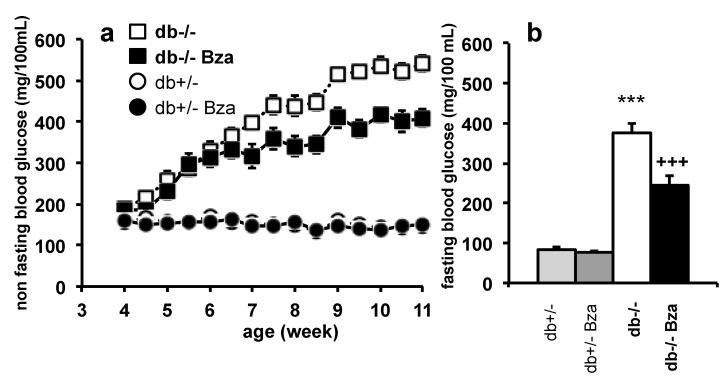
Influence of benzylamine supplementation on non-fasting and fasting blood glucose in db-/- and db+/- mice. Littermates from inbred db+/- mating were separated according to gender and distributed in a group subjected to benzylamine (Bza) at 0.5% in the drinking water (dark symbols) and in a control group with free access to water (open symbols) at week 4. Then at week 6, animals were separated according to their phenotype obese (squares, bold) or lean (circles, plain). (**a**) Non-fasting blood glucose as determined twice a week during 7-week treatment. Statistical analysis is reported in the body text of Results. (**b**) Overnight fasting blood glucose at the end of experiment. Data are means ± SEM of the following numbers of mice: 28 db-/-, 18 Bza-drinking db-/-, 21 db+/-, and 14 Bza-drinking db+/-. Significant difference between db-/- and db+/- at: *** *p* < 0.001; significant difference between Bza-db-/- and db-/- at: +++ *p* <0.001.

**Figure 2 nutrients-13-02622-f002:**
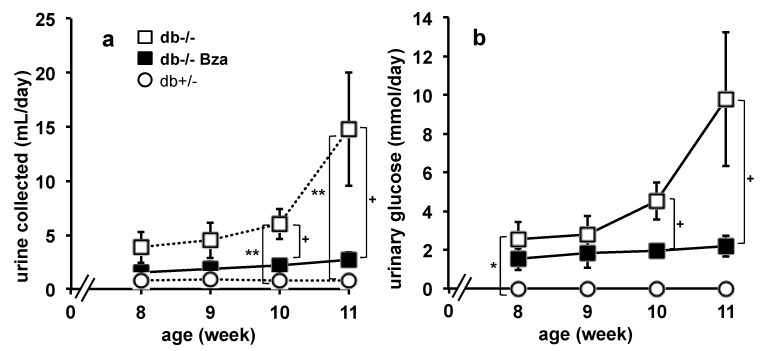
Benzylamine supplementation lowers urine production and glucose urinary output in obese/diabetic mice. (**a**) Daily urine emission of db-/- (open squares), Bza-drinking db-/- (closed squares) and db+/- mice (open circles) at the age of 8–11 weeks, once isolated in metabolic cages for 24 h. (**b**) Daily urinary glucose output. Mean ± SEM of the following numbers of mice, with the ratio of males/females indicated in parentheses: 11 (5/6) for db-/-; 9 (5/4) for Bza-db-/-; 10 (5/5) for db+/-. Significant influence of the genotype at: * *p* < 0.05, ** *p* < 0.01 for db-/- vs. db+/-; significant difference at: + *p* < 0.05 for db-/- vs. Bza-db-/- mice.

**Figure 3 nutrients-13-02622-f003:**
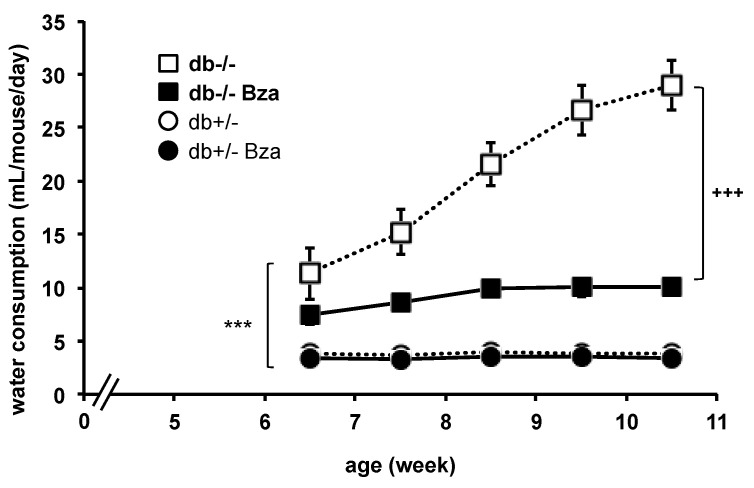
Benzylamine supplementation lowers water consumption in obese/diabetic mice. Each point is the mean ± SEM of *n* = 12 (7 males/5 females) for db-/- (open squares); 8 (4/4) for Bza-db-/- (closed squares); 9 (6/3) for db+/- (open circles); 5 (2/3) for Bza-db+/- (closed circles). Significant influence of genotype at: *** *p* < 0.001 for db-/- vs. db+/-; significant difference at: +++ *p* < 0.001 between db-/- and Bza-db-/- mice.

**Figure 4 nutrients-13-02622-f004:**
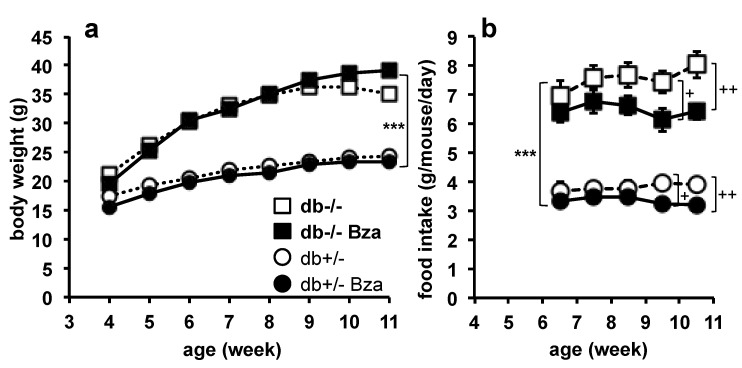
Influence of benzylamine supplementation on body weight gain and food intake in db-/- and db+/- mice. (**a**) Body weight of db-/- (squares) and db+/- mice (circles) drinking water (open symbols) or Bza 0.5% (closed symbols). (**b**) Average daily food intake. Mean ± SEM of *n* = 28 (17 males/11 females) for db-/-; 18 (9/9) for Bza-drinking db-/-; 21 (12/9) for db+/-; 14 (7/7) for Bza-drinking db+/- mice. From the age of 6 weeks, difference according to the genotype was significant at: *** *p* < 0.01. Significant difference at: + *p* < 0.05; ++ *p* < 0.01 for db-/- vs. Bza-db-/-, and for for db+/- vs. Bza-db+/- mice.

**Figure 5 nutrients-13-02622-f005:**
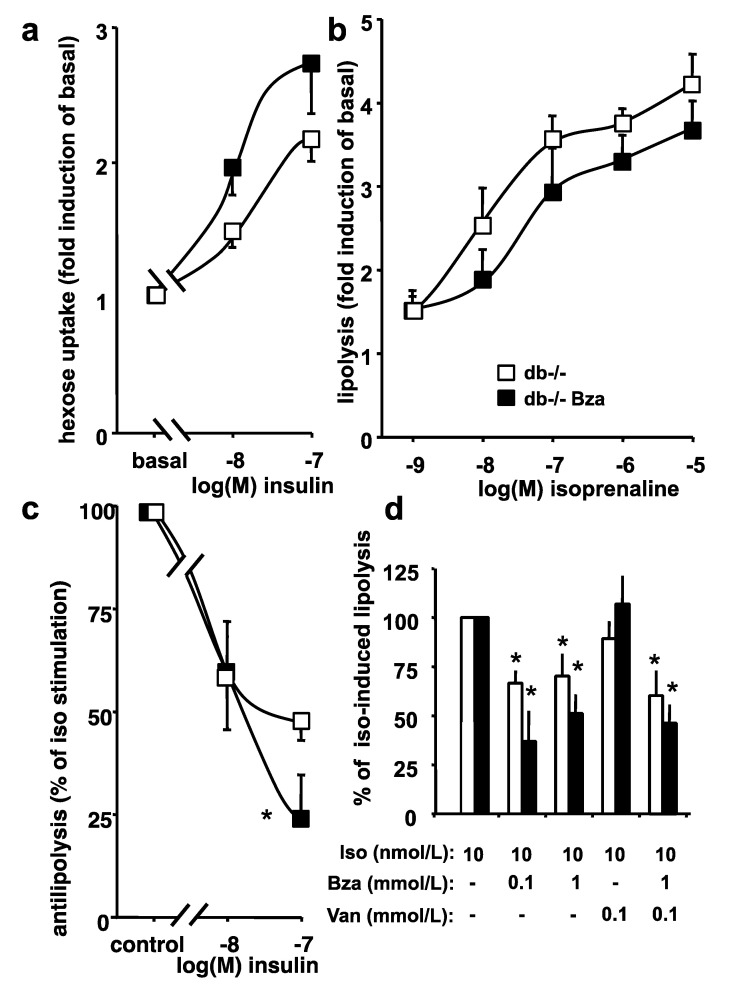
Influence of benzylamine supplementation on glucose transport and lipolytic activities in adipocytes from obese/diabetic mice. (**a**) Hexose uptake in response to insulin, relative to basal [^3^H]-2-deoxyglucose uptake in control (open squares) and Bza-drinking db-/- mice (closed squares). (**b**) Dose-dependent isoprenaline stimulation of lipolysis expressed as increase over basal glycerol release. (**c**) Inhibition by insulin stimulated lipolysis, expressed as percentage of the stimulation induced by 10 nmol/L of the β-adrenergic agonist isoprenaline. Difference significant at: * *p* < 0.05 for db-/- vs. Bza-drinking db-/-. (**d**) Inhibition by benzylamine (Bza) alone or in the presence of vanadium (Van) of isoprenaline-stimulated lipolysis (Iso), expressed as percentage of glycerol release. Different from 10 nmol/L isoprenaline alone at: * *p* < 0.05. Means ± S.E.M. of 4–6 adipocyte preparations/group.

**Figure 6 nutrients-13-02622-f006:**
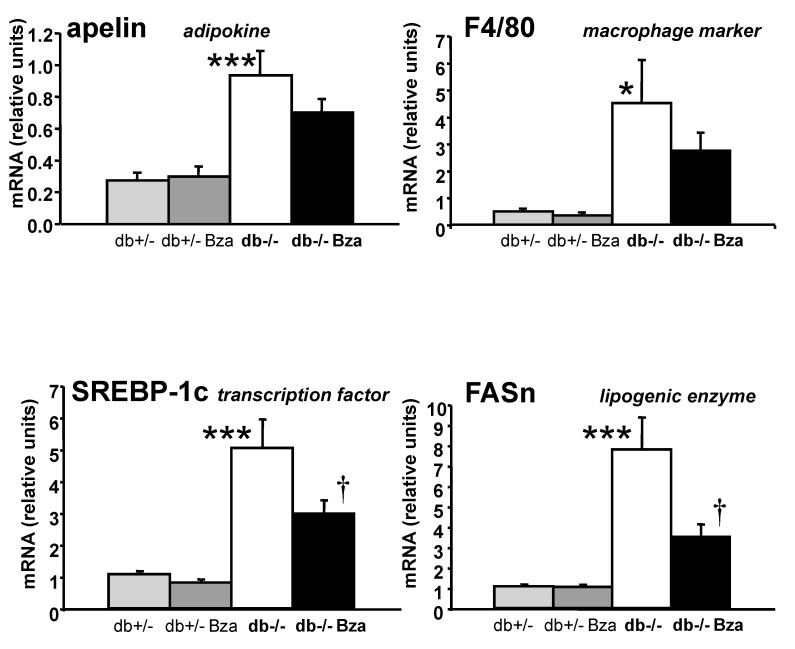
Normalization by benzylamine supplementation of the altered expression of several fuel metabolism- and inflammation-related genes in obese/diabetic mice. Upper panel: mRNA abundance of the adipokine apelin and of the macrophage marker F4/80 in SCWAT. mRNA abundance of the transcription factor SREBP-1C and of the lipogenic enzyme FASn in liver. Means ± SEM of 7–10 mice/group. Influence of genotype significant at: * *p* < 0.05; *** *p* < 0.001 for db-/- vs. db+/-. Significant difference between db-/- and Bza-treated db-/- at: † *p* < 0.05.

**Figure 7 nutrients-13-02622-f007:**
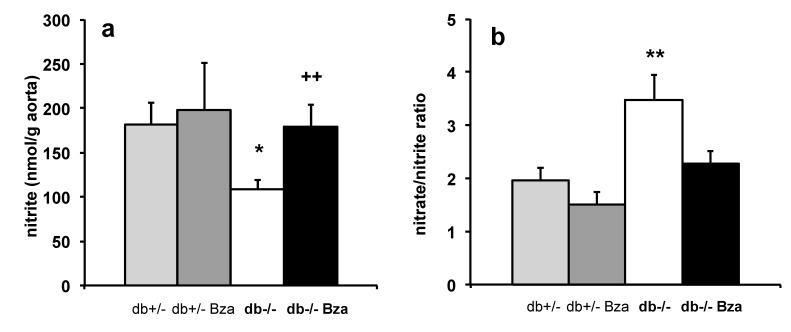
Nitrite levels (**a**) and nitrate/nitrite ratio (**b**) in aorta of db+/- (light grey), db-/- (white) and Bza-treaded db+/- (dark grey) and db-/- (black) mice. Data are means ± SEM of 5-14 determinations; significant difference at * *p* < 0.05 and ** *p* < 0.01 for db-/- vs. db+/-; ++ *p* < 0.01 for db-/- vs. Bza-db-/-.

**Table 1 nutrients-13-02622-t001:** Primer sequences for the studied mouse genes.

Mouse Gene	Oligonucleotide Sense
	/Antisense
*Adiponectin*	TGGAATGACAGGAGCTGAAGG TATAAGCGGCTTCTCCAGGCT
*Aoc3*	GTGGTCAGATCCGTGTCTACCTT CCTGTGGCGTGGAATTTGA
*Apelin*	TCTTGGCTCTTCCCTCTTTTCA GTGCTGGAATCCACTGGAGAA
*FASn*	ATCCTGGAACGAGAACACGATCT AGAGACGTGTCACTCCTGGACTT
*F4/80*	TGACAACCAGACGGCTTGTG GCAGGCGAGGAAAAGATAGTGT
*CD31*	GTCGTCCATGTCCCGAGAA GCACAGGACTCTCGCAATCC
*HSL*	GGCTTACTGGGCACAGATACCT CTGAAGGCTCTGAGTTGCTCAA
*PAI-1*	TCTCCAATTACTGGGTGAGTCAGA GCAGCCGGAAATGACACAT
*Tie2*	CAATCAGGCCTGGAAATACATTG TCCGCGGCTCCAAGTAGTT

**Table 2 nutrients-13-02622-t002:** Effect of Bza supplementation on adipose tissue and liver wet mass in 11-week old db+/- and db-/- mice.

Parameter	db+/-	Bza-db+/-	db-/-		Bza-db-/-	
body mass (g)	21.7 ± 0.8	21.5 ± 0.7	32.9 ± 1.0	***	36.2 ± 1.0	+
body weight gain (g)	7.6 ± 0.7	8. 7 ± 0.6	14.6 ± 1.1	***	19.7 ± 1.1	++
INWAT mass(g)	0.45 ± 0.02	0.42 ± 0.06	1.93 ± 0.10	***	2.24 ± 0.11	
SCWAT mass (g)	0.35 ± 0.03	0.29 ± 0.04	2.66 ± 0.18	***	3.31 ± 0.16	+
liver mass (g)	0.89 ± 0.03	0.91 ± 0.04	1.71 ± 0.07	***	1.87 ± 0.07	

As in Figure 1, results are means ± SEM of 14 mice for Bza-db+/-, and 18 for Bza-db-/- treated groups, (7/7 and 9/9 males/females, respectively), and 21–28 mice in their corresponding control groups db+/- and db-/-. Difference between db+/- and db-/- genotypes significant at: *** *p* < 0.001; difference between Bza-db-/- and db-/- significant at: + *p* < 0.05; ++ *p* < 0.01. No significant difference was found between db+/- and Bza-db+/- by two-way ANOVA. INWAT, intra-abdominal white adipose tissues; SCWAT, subcutaneous white adipose tissues.

**Table 3 nutrients-13-02622-t003:** Effect of benzylamine supplementation on plasma metabolic parameters in db+/- and db-/- after overnight fasting.

Plasma Level	db+/-	Bza-db+/-	db-/-		Bza-db-/-	
glucose (g/L)	1.10 ± 0.08	1.07 ± 0.13	5.67 ± 0.49	***	3.30 ± 0.43	++
insulin (μg/L)	0.40 ± 0.05	0.38 ± 0.05	2.89 ± 0.32	***	2.62 ± 0.51	
relative murine HOMA-IR	1.02 ± 0.16	0.76 ± 0.10	37.38 ± 5.48	***	20.69 ± 5.38	+
cholesterol (mmol/L)	1.94 ± 0.13	2.00 ± 0.13	2.29 ± 0.22	*	2.46 ± 0.19	
LDL cholesterol (mmol/L)	0.18 ± 0.04	0.20 ± 0.04	0.28 ± 0.12		0.23 ± 0.10	
HDL cholesterol (mmol/L)	1.62 ± 0.11	1.69 ± 0.11	2.01 ± 0.16	**	2.08 ± 0.15	
FFA (mmol/L)	0.64 ± 0.05	0.78 ± 0.07	0.90 ± 0.05	***	0.94 ± 0.06	
TG (g/L)	0.97 ± 0.07	1.13 ± 0.09	1.27 ± 0.10		0.87 ± 0.06	+
uric acid (µmol/L)	279.3 ± 18.0	272.6 ± 17.9	406.7 ± 23.5	***	330.6 ± 24.5	+
fructosamine (µmol/L)	204.5 ± 4.7	189.1 ± 7.7	291.8 ± 11.2	***	262.7 ± 14.3	

Number of mice per group, *n*: db+/-, 21; Bza-db+/-, 14; db-/-, 28; Bza-db-/-, 18. Means ± SEM were compared by two-way ANOVA. Significantly different from db+/- at: * *p* < 0.05; ** *p* < 0.01; *** *p* < 0.001. Significantly different from db-/- at: + *p* < 0.05; ++ *p* < 0.01. No significant difference was found between db+/- and Bza-db+/-. FFA, free fatty acids; TG, triacylglycerols; L/HDL, low/high density lipoproteins.

**Table 4 nutrients-13-02622-t004:** Altered gene expressions in SCWAT from obese mice that were not improved by oral benzylamine.

Function	Product	db+/-	db-/-		Bza-db-/-
Adipokine secretion	adiponectin	207.03 ± 16.36	298.20 ± 32.30	*	302.33 ± 21.49
	apelin	0.27 ± 0.05	0.93 ± 0.15	***	0.70 ± 0.08
Fatty acid metabolism	FASn	220.40 ± 43.23	77.47 ± 48.93	*	41.57 ± 4.88
	HSL	55.39 ± 4.06	86.08 ± 8.57	**	88.26 ± 6.16
Inflammation	PAI-1	2.04 ± 0.66	20.43 ± 4.14	***	26.59 ± 4.21
	F4/80	0.49 ± 0.08	4.51 ± 1.60	*	2.74 ± 0.68
Endothelial markers	CD 31	31.52 ± 3.25	43.85 ± 4.78	*	41.22 ± 3.09
	Tie 2	5.37 ± 0.62	8.06 ± 0.99	*	6.08 ± 0.85
Adipogenesis marker	AOC3	30.68 ± 3.37	78.22 ± 7.58	***	84.64 ± 7.10

Relative mRNA abundance given as mean ± SEM of 9 mice. Significantly different from db+/- at: * *p* < 0.05; ** *p* < 0.01; *** *p* < 0.001. No significant difference was found between db-/- and Bza-db-/-.

**Table 5 nutrients-13-02622-t005:** Hydrogen peroxide release in SCWAT homogenates from mice subjected or not to benzylamine supplementation.

H_2_O_2_ Release (pmol/mg prot/min)	db+/-	Bza-db+/-	db-/-	Bza-db-/-
basal	65 ± 13	102 ± 24	113 ± 19 *	103 ± 9
AOC3-dependent oxidation of Bza 0.1 mmol/L	152 ± 10	171 ± 29	299 ± 47 *	252 ± 37

Means ± SEM of 14 mice for Bza-db+/-, and 18 for Bza-db-/- treated groups, (7/7 and 9/9 males/females, respectively), and 21-28 mice in their corresponding control groups. Difference between db+/- and db-/- genotypes significant at: * *p* < 0.05. No significant difference was found between control and Bza-supplemented groups.

## Data Availability

No additional data is available.

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
