# Peer review of "Oral Supplementation with Benzylamine Delays the Onset of Diabetes in Obese and Diabetic db-/- Mice"

_nutrients, 2021, doi:10.3390/nu13082622_

Round 1
Reviewer 1 Report
The authors conduct an animal model study and aimed to investigate the oral supplementation with benzylamine delays the onset of diabetes in obese and diabetic db-/- Mice.
Comments:
1.
Different sample sizes are examined.??
Figures 1 and 4: n = 28 db -/-, 18 Bza-drinking db -/-, 21 db +/-, and 14 Bza-drinking db +/-
Figure 2: n = 11 (5/6) for db -/-; 9 (5/4) for Bza-db -/-; 10 (5/5) for db +/-.
Figure 3: n = 12 (7 males/5 females) for db -/- ; 8 (4/4) for Bza-db -/-; 9 (6/3) for db +/- ; 5 (2/3) for Bza-db +/-
Figure 5:
(a) to (c), db -/- vs Bza-drinking db -/-, n??; (d) n= 4-6 adipocyte preparations/group
Figure 6: n=7-10 mice/group
Figure 7: n=5-14 determinations
Tables 1, 2 and 4: n = 28 db -/-, 18 Bza-drinking db -/-, 21 db +/-, and 14 Bza-drinking db +/-
Table 3. n= 9 mice
It might have a selection bias (recruited mice) in this animal model study.
2.
Adding a graphical summary or graphical abstract of the study design, maybe it is easy to understand the principal message of the paper very quickly.
Author Response
The authors conduct an animal model study and aimed to investigate the oral supplementation with benzylamine delays the onset of diabetes in obese and diabetic db-/- Mice.
Thank you, your comment is absolutely right.
Comments:
- Different sample sizes are examined.??
Figures 1 and 4: n = 28 db -/-, 18 Bza-drinking db -/-, 21 db +/-, and 14 Bza-drinking db +/-
Figure 2: n = 11 (5/6) for db -/-; 9 (5/4) for Bza-db -/-; 10 (5/5) for db +/-.
Figure 3: n = 12 (7 males/5 females) for db -/- ; 8 (4/4) for Bza-db -/-; 9 (6/3) for db +/- ; 5 (2/3) for Bza-db +/-
Figure 5: (a) to (c), db -/- vs Bza-drinking db -/-, n??; (d) n= 4-6 adipocyte preparations/group
Figure 6: n=7-10 mice/group
Figure 7: n=5-14 determinations
Tables 1, 2 and 4: n = 28 db -/-, 18 Bza-drinking db -/-, 21 db +/-, and 14 Bza-drinking db +/-
Table 3. n= 9 mice
It might have a selection bias (recruited mice) in this animal model study.
Answering to such reviewer's comments is a hard task for co-authors because we thought that we clearly exposed in the Materials and Methods section why the number of mice was not the same among the different studied groups and why it could vary from an exploration to another. Aware of this fact, we never have omitted to mention the sample size for all the measurements reported in Results. However, if the readers jump directly to Results without paying attention to the explanations given in Materials and Methods section, it will be difficult for us to describe in a clearer manner the experiments we performed. We cannot copy/paste here in these point-by-point answers to comments the entire section 2.2 " Group composition and oral benzylamine supplementation", but we can ascertain that it contains all the response to the raised concerns. In this section, we justify that we compared littermates obtained from the crossing of heterozygous +/- db/db mice. Really, all the necessary details for the description of the control and treated groups compared under strictly parallel conditions are given in the Methods section and throughout the Ms, including in the revised version. The Mendelian frequency was correct among the studied littermates, being in the offspring close to 1/4 for obese db -/-, 1/4 for lean db +/+, and 1/2 for normoglycemic db +/-. However, the number of animals was not the same for each of the litter studied and the gender ratio was also slightly varied from one litter to another. In the original section 2.2, we already indicated that benzylamine supplementation started at weaning (4-week old) i.e. at the time of gender separation, but before knowing whether the mice will become obese or lean, a phenotyping trait determined at six weeks of age. Although we cannot present below an exhaustive list of the mice used, we can briefly summarize here that the maximal size of the groups resulting from separation according to phenotype reached: 28 obese db -/- and 21 lean db +/- (both genders), for constituting the control groups while, among the littermates, there was 18 future obese db -/- and 21 lean db +/- (both genders) subjected to benzylamine treatment As noted by the reviewer, this group size is common for figs 1-4 and for the three tables (body weight, metabolic parameters, etc) while lower number of animals was studied for other outcomes. This was a consequence of limited biological resource (as already mentioned in the original Materials and Methods). For instance, when adipose samples were subjected to mRNA quantification, they could not be used for the exploration of insulin or benzylamine effects on functional adipocytes. Regarding a possible bias of such random distribution of the available material to measurements with different methodologies, we can attest we made all our efforts to avoid any selection within the compared groups. For the sake of concision, we have not added further related details in the Ms since the issues of biological resource limitation or of restricted access to MRI devices was already mentioned in various occurrencesinthe Ms, such as in the top of p 9, in which it is reported that the weighing of the dissected fat depots confirmed the increased percentage of body fat content, previously measured in a non-invasive manner on a smaller number of mice of the same group, after prior adaptation to the instrument.
- Adding a graphical summary or graphical abstract of the study design, maybe it is easy to understand the principal message of the paper very quickly.
We are sorry for the mismatch, as a graphical abstract was already uploaded in the MDPI platform during Ms submission. We hope that under its present form, the graphical abstract will be useful and accessible after revision.
Reviewer 2 Report
The article entitled “Oral Supplementation with Benzylamine Delays the Onset of 2 Diabetes in Obese and Diabetic db-/- Mice” explain the oral supplementation of Benzylamine in type 2 diabetic animals. The authors could have also focused on protein makers using western blots images and immune histochemical analysis which is lacking in this review article. Although I don't have major concerns with this article, minor issues need to be addressed appropriately.
Narratives
- Animal model section db gene (db +/-), genotypes needs to superscripted “+/-“ throughout the manuscript
- Group composition and oral benzylamine supplementation section db -/- mice, genotypes to be “-/-“ needs to superscripted throughout the manuscript
- Group composition and oral benzylamine supplementation section genotypes db +/+, “+/+” to be superscripted
- The sequences of validated primers to be listed as a table in this manuscript
- Figure 1a, the authors have to label the groups in the graph also, not only on the figure legend
- Figure 2, the authors have to label the groups in the graph also, not only on the figure legend
- Figure 3, the authors have to label the groups in the graph also, not only on the figure legend
- Above comment applies to figure 4
- Table 1, all the significant to be superscripted
- Above comment applies to Table 2
- HOMA-IR needs to be calculated and presented
- Table 3, significance to be superscripted
- Figure 7, serum Total nitrite levels to be added

Author Response
The article entitled “Oral Supplementation with Benzylamine Delays the Onset of 2 Diabetes in Obese and Diabetic db-/- Mice” explain the oral supplementation of Benzylamine in type 2 diabetic animals. The authors could have also focused on protein makers using western blots images and immune histochemical analysis which is lacking in this review article. Although I don't have major concerns with this article, minor issues need to be addressed appropriately.
Thank you for your careful perusal. We agree with referee's remarks, but we have to repeat here that the major outcome of benzylamine supplementation was the limitation of hyperglycaemia and of glycosuria, two biochemical parameters that do not deserve western blotting or histochemistry for being displayed.
Narratives
Animal model section db gene (db +/-), genotypes needs to superscripted “+/-“ throughout the manuscript
Group composition and oral benzylamine supplementation section db -/- mice, genotypes to be “-/-“ needs to superscripted throughout the manuscript
Group composition and oral benzylamine supplementation section genotypes db +/+, “+/+” to be superscripted
Thank you for this suggestion. We respectfully agree with the nomenclature using superscript for the homozygous and heterozygous mice used in this study, as proposed by the reviewer. Nevertheless, modifying our article as requested will results in almost illegible labels for the model used, especially in the legends of figures, for which the final font size used by MDPI is "palatino linotype 9". Can the reviewer consider that the difference between obese and lean littermates will be summarized to several pixels only in the case of using superscript with automatically reduced size, rendering therefore difficult the perusal for numerous readers (-/- and +/- instead of -/- and +/-). As an open access article, once published this work may be visualized / printed worldwide by computer screens / printers with sometimes poor definition, making quite mandatory the use of enlargement. Taking these two components in consideration, all the co-authors have therefore decided to use only once the superscript for the first description of the mice (also named C57BL/KsJ mice) in "Introduction" and "Animal model " section , then to use our chosen nomenclature throughout the Ms, after a complete presentation of the three genotypes made in the last line of this section: " ... obese and diabetic db -/-, normoglycemic db +/-, and lean db +/+". This can be considered as correct regarding the rules of using abbreviations, since introduced at their first occurrence in Results. Please note that all the modifications have been performed in red font in the text of the revised version.
The sequences of validated primers to be listed as a table in this manuscript
Thank you for this suggestion. We have added a novel Table 1 for answering to that request and renumbered all other tables.
Figure 1a, the authors have to label the groups in the graph also, not only on the figure legend
Figure 2, the authors have to label the groups in the graph also, not only on the figure legend
Figure 3, the authors have to label the groups in the graph also, not only on the figure legend
Above comment applies to figure 4.
All these suggestions have been taken into account: Thanks for helping us in improving the understanding of the figures.
Table 1, all the significant to be superscripted
Above comment applies to Table 2
Table 3, significance to be superscripted
All these suggestions have been taken into account: the modifications appear in red in the revised version, while all co-authors disagree with the resulting lowered readability that clearly does not improve the original version.
HOMA-IR needs to be calculated and presented
We respectfully agree with your suggestion. However we did not used this parameter in the original Ms for the following reason: the homeostatic model assessment is a method used to quantify insulin resistance and beta-cell function (HOMA-IR) in humans. While it is of clinical interest, the mere calculation of [glucose mM]x[insulin µU/mL]/22.5 is out of its limits of relevance for the model of severe insulin-resistant diabetes that is the db/db mouse. Even the blood glucose of >5g/L of this model is overwhelming for diabetic patients. However, as this surrogate measure of insulin resistance can been used in rodents with caution/adaptations (doi:10.1152/ajpendo.90889.2008), we can answer positively to the reviewer request by adding in novel Table 3 the values of such relative HOMA-IR, which are about twice higher in untreated diabetic mice when compared to those receiving benzylamine. This additional outcome is useful for improving our message.
Figure 7, serum Total nitrite levels to be added
Unfortunately, we spared sufficient plasma samples from 4 obese mice only for such supplemental parameter, which is insufficient to further support the improvement of the oxidative metabolism of NO as assessed by the ratio of nitrate to nitrite in aorta.
Round 2
Reviewer 1 Report
No further comment